# The Effect of Signal Duration on the Classification of Heart Sounds: A Deep Learning Approach

**DOI:** 10.3390/s22062261

**Published:** 2022-03-15

**Authors:** Xinqi Bao, Yujia Xu, Ernest Nlandu Kamavuako

**Affiliations:** 1Department of Engineering, King’s College London, London WC2R 2LS, UK; xinqi.bao@kcl.ac.uk (X.B.); yujia.xu@kcl.ac.uk (Y.X.); 2Faculté de Médecine, Université de Kindu, Kindu, Maniema, Democratic Republic of the Congo

**Keywords:** heart sound, deep learning (DL), recurrent neural networks (RNNs), convolutional neural network (CNN)

## Abstract

Deep learning techniques are the future trend for designing heart sound classification methods, making conventional heart sound segmentation dispensable. However, despite using fixed signal duration for training, no study has assessed its effect on the final performance in detail. Therefore, this study aims at analysing the duration effect on the commonly used deep learning methods to provide insight for future studies in data processing, classifier, and feature selection. The results of this study revealed that (1) very short heart sound signal duration (1 s) weakens the performance of Recurrent Neural Networks (RNNs), whereas no apparent decrease in the tested Convolutional Neural Network (CNN) model was found. (2) RNN outperformed CNN using Mel-frequency cepstrum coefficients (MFCCs) as features. There was no difference between RNN models (LSTM, BiLSTM, GRU, or BiGRU). (3) Adding dynamic information (∆ and ∆²MFCCs) of the heart sound as a feature did not improve the RNNs’ performance, and the improvement on CNN was also minimal (≤2.5% in *MAcc*). The findings provided a theoretical basis for further heart sound classification using deep learning techniques when selecting the input length.

## 1. Introduction

Cardiovascular diseases (CVDs) are the leading cause of death worldwide, accounting for approximately 31% of the mortality [1]. Auscultation is the most common and effective way in early screening, which plays a vital role in CVD detection for necessary action to lower the risk for worsening heart diseases. However, it relies significantly on the physician’s listening ability and clinical experience. Experienced cardiologists can distinguish between 73% and 80% accuracy of pathological murmurs, while inexperienced new physicians or trainees can distinguish 20–40% accuracy [2,3]. A misdiagnosis will lead the patient to miss the best time for treatment or increase the cost due to unnecessary further detection (e.g., electrocardiogram, cardiac ultrasound, and computerised tomography).

Heart sound is produced by the closure of a valve or tensing of a chordae tendineae that the physician will listen to during the auscultation. The acoustic sensors can capture it, and its waveform are visualised as a phonocardiogram (PCG), as shown in Figure 1. In healthy conditions, S1 and S2 are the two main components in the waveform, representing the sound of mitral and tricuspid valve closure (S1) and the closure of the aortic and pulmonic valves (S2), respectively. In addition, S3 and S4 are also innocent components seen on children’s PCG but rarely seen in adults. They indicate the sound caused by an increase in ventricular blood volume and an atrial gallop by blood being forced into a stiff ventricle. Pathological heart sounds differ from healthy ones due to the murmurs primarily caused by the abnormal heart structure. Murmurs can occur in the systolic interval or the diastolic interval, reflecting different types of CVDs, e.g., the most common mitral or aortic stenosis murmurs can be seen during systole [4]. As shown in Table 1, the heart sound components are short in time and low in frequency, and its principal frequencies are at the lower end of the human ear (20–20k Hz), which is not sensitive to hear. Therefore, traditional auscultation has natural shortcomings.

Computer-aided heart sound analysis has great potential to improve auscultation accuracy by overcoming human hearing limitations and clinical experience. An electronic stethoscope is used instead of the traditional acoustic stethoscope to record the heart sound signal. Then, a machine learning classification algorithm will conduct the automated diagnosis. Typical PCG signal classification algorithms include three significant steps: segmentation (including component identification), feature extraction, and classification [5]. Firstly, the conventional segmentation breaks the whole signal into each heart cycle and locates the heart sound components, as shown in Figure 1. It should be noted that the meaning of segmentation in the heart sound processing is slightly different from the typical signal processing. In standard signal processing, segmentation means cutting the signal into segments by moving windows.

In contrast, heart sound processing means breaking the signal into heart cycles and indicating the heart sound components. To avoid confusion, the ‘segmentation’ in this paper means the conventional heart sound segmentation, and we will use other verbs to describe our signal processing. After segmentation, features such as time-frequency, energy-based, wavelet transform, and Mel-frequency cepstrum coefficients (MFCCs) are extracted from the signals to train a classifier. MFCCs are the most commonly used features in sound processing studies. Their frequency bands in Mel-scale can approximate human auditory system response more closely than linear-scaled spectrums because they consider the human ear perception sensitivity concerning the changing frequencies. Thus, using MFCCs as features is particularly suitable for simulating the auscultation activity. Afterwards, machine learning methods will classify the input heart sounds into normal and abnormal classes based on these features.

It is still an open question on the necessity of segmentation. As mentioned, conventional segmentation aims to locate the heart sound components, helping extract the features, especially in the time domain. However, there is no widely recognised golden PCG segmentation technique. Unsuccessful segmentation will conversely affect the accuracy and robustness of the algorithm and increase computation load. In addition, several representative studies without conventional segmentation have achieved considerable performance in the heart sound classification [6,7,8,9,10,11,12]. Thus, segmentation becomes an optional step that should depend on the selected features and classifiers. Generally, algorithms without segmentation make use of deep learning methods based on artificial neural networks (ANNs) such as convolutional neural networks (CNNs) and recurrent neural networks (RNNs). The signal is not segmented to identify components in these algorithms but broken into signal pieces of equal duration. As a result, different studies cut the heart sound recordings into various epoch durations, e.g., some studies broke the recordings into one-second epochs to enlarge the dataset as much as possible for training and validation [9,12]. In contrast, others were cut into five-second pieces to retain more information in each segment [10,13]. Different data structures and settings make it hard to evaluate the proposed methods, so it is necessary to unravel how to choose the appropriate PCG input duration for a given dataset. 

The aim of this study is threefold: (1) to systematically investigate how the duration of input PCG signals will affect the performance of deep learning methods; (2) to compare the performance of commonly used RNNs, including gated recurrent unit (GRU), long short-term memory (LSTM), Bidirectional LSTM, and GRU (BiLSTM and BiGRU) under different epoch lengths; (3) to ascertain if adding dynamic information (deltas and delta-deltas of MFCCs) as the feature can improve the performance of the tested deep learning methods. Ultimately, the findings of this study will provide insight to determine reasonable input length, classifiers, and features for designing deep learning PCG classification algorithms in the future.

## 2. Related Works

The earliest publication on automated PCG classification can be traced to 1963, when Gerbarg et al. used a threshold-based method to identify rheumatic heart disease [14]. Since then, many articles have been published on the PCG segmentation techniques, features selection, and classification methods. The proposed classification algorithms include logistic regression [15], random forest [16], K-nearest neighbours (KNN) [15,17,18], regression tree [19], support vector machine (SVM) [15,20,21], hidden Markov model (HMM) [22], and ANN and its variants [23,24,25]. However, it was almost impossible to systematically and uniformly evaluate and compare the early research performance in this field, as they used different datasets with different classification tasks. With the development of deep learning techniques in recent years, more researchers switched from traditional machine learning to deep learning methods to design improved classification algorithms.

In 2016, PhysioNet/Computing in Cardiology (CinC) Challenge created an extensive database sourcing from nine different heart sound databases with 4430 recordings collected from 1072 healthy participants and patients with a variety of conditions [26]. This database can be used for binary classification (normal and abnormal heart sound) and as the platform to assess classifiers and features objectively. Furthermore, it provided the basis for designing deep learning PCG classification methods. Despite ambiguous discussion on the best classifier or the selection of features according to the PhysioNet challenge results, almost half of the top performances, especially the top three applied the neural network algorithms. This showed a great potential of deep learning in improving the performance of automated PCG classification. For the feature selection, 4 of the top 10 algorithms used MFCCs, given the proven track of their reliability and universality [27].

Moreover, Yang and Hsieh used RNN without segmentation, ranking 13 in the Challenge (*Acc*: 79%) [28]. This method aroused interest in the necessity of segmentation and led to future research on PCG classification using RNNs models. Practically, RNNs are suitable for sound recognition with the advantage of exhibiting temporal dynamic characteristics, and it has achieved great success in automatic handwriting and speech recognition [29]. After the challenge, more researchers explored deep learning methods without segmentation using the PhysioNet database. Table 2 outlines several representative studies in recent years.

From Table 2, MFCCs could be considered the most frequent and mainstream recognised feature. It is still controversial whether CNN or RNN (LSTM, BiLSTM, and GRU) is more suitable for heart sound classification. A combined classifier seems to be the future trend that could contain the advantages of both CNN and RNNs. Because the data length of the recordings in the PhysioNet database varies from 5 s to 2 min, different studies have cut the raw data into smaller lengths without segmentation. They have then rebuilt the pre-processed datasets for training, validation, and testing. This unifies the data, enlarges the training datasets, and improve the classification performance to a certain extent. Typically, the epoch durations ranged from 1 s (approximately one heart cycle) to 5 s (shortest heart sound recording in the PhysioNet database). However, various data structures make it hard to compare their claimed performance objectively. Moreover, it is unclear if the best input duration can ensure that both training sample size and each sample’s information are sufficient.

This paper will fill the mentioned knowledge gap by breaking the PhysioNet database into different lengths (1–5 s) to rebuild new datasets. Inspired by the previous studies, MFCCs are chosen as features. Commonly used CNN and RNNs structures will be compared under similar settings. The study will assess the effect of input length and classifier selection on model performance.

## 3. Proposed Classification Algorithms

This section will introduce the concept of MFCC-based features, including standard MFCCs, the first-order MFCC (∆MFCCs), and the second-order MFCC (∆²MFCCs). A detailed description of the neural network models used in this study, including CNN, LSTM, BiLSTM, GRU, and BiGRU, will follow.

### 3.1. Extraction of MFCC Features

Mel-frequency cepstrum (MFC) is a vital representation in sound processing that uses Mel-scale instead of linear scale to display the short-term power spectrum. The advantage of the Mel-scale is that it can reflect more closely to the human non-linear auditory system. The human auditory sense is more sensitive to identifying the voice changes in low frequency. In contrast, people need a larger frequency band in high frequency (over 500 Hz) to distinguish the same pitch increment. The relationship between the two scales (Hz and Mel) is:(1)$$Mel\left(f\right)=2595log_{10}\;\left(1+\frac{f}{700}\right)$$
where *Mel*(*f*) is the frequency in the Mel-scale, *f* is the frequency in the linear scale. MFCCs are the coefficients derived at Mel filter banks that describe the energy distributed in the different critical frequency bands. The extraction of MFCCs includes the following steps:

(1)Pre-emphasis

The purpose of pre-emphasis is to amplify and compensate the high-frequency component that is suppressed in sound production. This will increase the Signal-to-Noise Ratio (SNR) and balance the frequency by enhancing the high-frequency content, which is usually small in magnitudes. A pre-emphasis filter shown in Equation (2) is generally applied.
(2)x(t)=s(t)−μs(t−1) 
where *s*(*t*) is the sound signal, *x*(*t*) is the filtered signal, and *μ* ∊ [0.9, 1] is used for high-pass pre-emphasis.

(2)Fourier transform

To obtain the spectrum of the input signal, a short-time Fourier transform is used to convert the signal from the time domain into the frequency domain. The conversion is:(3)$$X\left(k\right)=\overset{N-1}{\underset{n=0}{\sum }}x\left(n\right)e^{-2j\pi nk/N},\;0\leq k\leq N$$
where *x*(*n*) is the input signal, *X*(*k*) is the corresponding Fourier coefficients, and *N* represents the number of samples in each frame. The power spectrum is squared $X{\left(k\right)}^{2}$.

(3)Mel-scaled power spectrum

Mel spectrum could be obtained by the power spectrum passing through a set of Mel-scaled filter banks, where the banks $B_{m}\left(k\right)$ is:(4)$$B_{m}\left(k\right)=\{\begin{array}{l} 0,\;k<f\left(m-1\right)\\ \frac{k-f\left(m-1\right)}{f\left(m\right)-f\left(m-1\right)},\;f\left(m-1\right)\leq k<f\left(m\right)\\ \frac{f\left(m+1\right)-k}{f\left(m+1\right)-f\left(m\right)},f\left(m\right)\leq k\leq f\left(m+1\right)\\ 0,\;\;\;\;k>f\left(m+1\right) \end{array}$$

*m* is each Mel filter of all M filters. *k* is the samples in frames. *f*(*m* − 1), *f*(*m*) and *f*(*m* + 1) represent the beginning, medium, and end frequency of each Mel triangle filter. The Mel-scaled power spectrum is the product of the power spectrum $X{\left(k\right)}^{2}$ and the banks $B_{m}\left(k\right)$, which is:(5)$$P\left(n\right)=\overset{N-1}{\underset{n=0}{\sum }}X{\left(k\right)}^{2}B_{m}\left(k\right),0\leq m\leq M$$

(4)Discrete cosine transform (DCT)

MFCCs could be derived by the DCT of the logarithmic Mel spectrum:(6)$$MFCCs\left(i\right)=\overset{M}{\underset{m=1}{\sum }}\log\left[P\left(n\right)\right]\cos\left[i\left(m-\frac{1}{2}\right)\frac{\pi }{M}\right],\;i=1,\;2,\;3,.\;.\;.,\;L$$
where L shows the order of frame for the MFCCs and *M* represents the number of Mel filter banks.

(5)Dynamic characteristic (∆ & ∆²MFCCs)

Besides standard MFCCs, we explore adding the ∆MFCCs and ∆²MFCCs as input features in this study. Because MFCCs can only describe the sound signal’s static information (spectral envelope shape), heart sound is inherently time-variant, and the dynamic information may help describe the signal more accurately. In addition, the human ear is more sensitive to sound changes. Thus, to better emulate the auscultation procedure, we add the dynamic information of the heart sound signal aiming to improve the detection accuracy. The extraction of ∆MFCCs is:(7)$$d_{i}=\frac{\sum_{n=1}^{N}n\left(MFCCs_{i+n}-MFCCs_{i-n}\right)}{2\sum_{n=1}^{N}n^{2}}$$
where $d_{i}$ is delta coefficient between $MFCCs_{i+n}$ and $MFCCs_{i-n}$ at frame *i*, and *N* is usually set to be 2. Similarly, ∆²MFCCs ($D_{i}$) could be calculated by:(8)$$D_{i}=\frac{\sum_{n=1}^{N}n\left(d_{i+n}-d_{i-n}\right)}{2\sum_{n=1}^{N}n^{2}}$$

### 3.2. Deep Learning Models

Deep learning (or deep neural networks, DNNs) refers to the machine learning algorithms using a neural network with more than one hidden layer. These interconnected layers are sequential, consisting of neurons to multiply their input and corresponding weight. The sum of the neural outputs is passed to the subsequent layer neurons after activation functions such as a sigmoid, tanh, or rectified linear unit (ReLU). This procedure is repeated until the final output layer, and the training procedure optimises the weights during forward and backward propagation. As indicated in Table 2, this has proven successful for heart sound classification. Commonly used DNNs include:

(1)Convolutional Neural Network (CNN)

CNN is a deep learning model that consists of convolutional (Conv), pooling, and fully connected layers. The Conv layer has a set of spatially small and learnable filters (kernels) working as feature detectors. They move across the input matrix “like sliding window”, calculating the dot product of the kernel parameters and windowed inputs. The output can be interpreted as the extracted feature map subject to the kernels. Usually, more than one kernel is used for the convolution. Thus, the output size will be several times larger than the input. For instance, an input size 32 × 32 (1024) matrix after four kernels (size 3 × 3, stride 1, padding 0) convolution will output 4 × 30 × 30 (3600). Therefore, the pooling layer is used for down-sampling to reduce the feature maps and the amount of computation and control overfitting. Max pooling is the most commonly used pooling layer, which remains the maximum in the rectangular filtered region. Following the last instance, a 2 × 2 size max-pooling filter with a stride of two will reserve only 25% of its original size, so only 800 parameters are transferred to the next layer. The fully connected layers can map the extracted features into catalogues for classification. Sometimes, a SoftMax layer follows the fully connected layer to the logits into a class probability distribution before the final output.

(2)Recurrent Neural Networks (RNNs)

Unlike CNN focusing on the spatial characteristics of the input, RNNs are specialised in processing sequence data such as time series, text, and audio. Generally, RNNs conduct the same computation procedure cyclically on each segment of the sequences, and the following output is based on previous calculations. From network structure, it includes memory to store the hidden internal state $h_{t}$, which could be calculated by the previous hidden state $h_{t-1}$ and input $x_{t}$, that
(9)$$h_{t}=f_{W}\left(W_{xh}x_{t}+W_{hh}h_{t-1}+b_{h}\right)$$
where $f_{W}$ refers to the hidden layer function such as a tanh activation function with parameter W shared across time (i.e., $W_{xh}$ indicates the weight of the input-hidden layer, $W_{hh}$ is the weight of the hidden-to-hidden layer, and $W_{yh}$ is the weight of hidden-to-output). b is the corresponding bias vector. The predicted output is:(10)$$y_{t}=W_{yh}h_{t}+b_{y}$$

However, this general architecture will face exploding weights and vanishing gradient issues on long-term sequences. Therefore, methods such as LSTM were proposed to improve this condition.

**Long Short-Term Memory (LSTM)**: LSTM reminds the general RNN architecture and changes the memory cell unit structure, making it capable of storing extended time intervals. As shown in Figure 2a, input, output, and forget gates control the information flow within the memory cell. For a given input $x_{t}$ at specific time *t*, the corresponding output after passing the gates are:(11)$$I_{t}=\sigma \left(W_{IF}x_{t}+W_{hI}h_{t-1}+W_{CI}C_{t}+b_{I}\right)\;O_{t}=\sigma \left(W_{xO}x_{t}+W_{hO}h_{t-1}+W_{CO}C_{t}+b_{O}\right)\;F_{t}=\sigma \left(W_{xF}x_{t}+W_{hF}h_{t-1}+W_{CF}C_{t}+b_{F}\right)$$
where *I*, *O*, and *F* represent the input, output, and forget gate, respectively. Similar to Equations (9) and (10), *W* is the weight of recurrent connections (i.e., $W_{IF}$ indicates the weight of the input-forget gates layer). *h* is the hidden state, and *b* is the bias. *σ* is the sigmoid activation function. The memory $C_{t}$ of this unit can be obtained by:(12)$$C_{t}=F_{t}C_{t-1}+I_{t}\tan h\left(W_{xC}x_{t}+W_{hC}h_{t-1}+b_{C}\right)$$

The hidden state vector (output vector) of this LSTM unit will be transferred to the next time interval, and it can be calculated by:(13)$$h_{t}=O_{t}\tan h\left(C_{t}\right)$$

**Gated Recurrent Unit (GRU)**: GRU can be regarded as a simplified LSTM unit that integrates the input gate with the forget gate, forming a new update gate to decide the acceptance or abandonment of the information. In addition, there is a reset gate to determine how much memory is to be forgotten. The two gates work together to adaptively remember or forget during the sequence reading. Its structure is shown in Figure 2b. The computational flow in one unit is given below:(14)$$\begin{array}{l} U_{t}=\sigma \left(W_{xU}x_{t}+W_{hU}h_{t-1}+b_{U}\right)\\ R_{t}=\sigma \left(W_{xR}x_{t}+W_{hR}h_{t-1}+b_{R}\right)\\ {\hat{h}}_{t}=\tan h\left(W_{\hat{h}x}x_{t}+U_{h{\hat{h}}_{t}}\left(R_{t}h_{t-1}\right)+b_{\hat{h}}\right)\\ h_{t}=\left(1-U_{t}\right){\hat{h}}_{t-1}+U_{t}{\hat{h}}_{t} \end{array}$$
where *U* and *R* represent the update and reset gate. $\hat{h}$ and *h* are the candidate activation vector and output vector. The nomenclature of the rest variables is the same as the equations before.

**Bidirectional Recurrent Neural Networks (BRNN)**: Standard RNNs are unidirectional with the constraint that they can only predict the current state based on previous information. Because future information is not reachable at that moment, bidirectional RNNs were proposed to improve this situation by connecting opposite directional hidden layers to the same input. As a result, the output layer can obtain both previous and future states information by the forward and backward pass. The structure of the BRNN is shown in Figure 2c. In this study, we will also explore using bidirectional LSTM and GRU to investigate if there can be an improvement in the detection accuracy.

## 4. Data Analysis

### 4.1. Datasets and Preprocessing

The database used in this study was the PhysioNet database, which consists of 3153 recordings, including 2488 normal and 665 abnormal cases. They were recorded by different research teams using different electronic stethoscopes under both clinical and non-clinical settings. Because of the uncontrolled measuring environment, the duration of the recordings ranged from 5 to 120 s. Different noise types such as body motion, ambient noise, and inside body sound (i.e., intestinal sound) were added to the original heart sound. This fits the actual auscultation situation but causes more difficulty to the classification algorithm. In addition, the subjects included children, adults, and the elderly. The abnormal cases involve various heart conditions, especially coronary heart disease and valvular diseases. More details about it can be seen in [26].

As the principal objective of this study was to investigate the PCG duration effect on the classification performance, the raw heart sound recordings in the database were further cut into 1 s (71,344 segments), 2 s (34,982 segments), 3 s (22,510 segments), 4 s (16,749 segments), and 5 s (13,015 segments) length without overlapping. Their labels were also generated according to the raw database into normal and abnormal (two classes). Since the testing datasets in the PhysioNet Challenge were not published, we divided the available datasets into training: validation: testing by 8:1:1.

### 4.2. Features Extraction

This study used MFCCs as the input feature for the deep learning models. The window framing for the extraction was a hamming window with 30 ms length and 20 ms overlap. Thirteen MFCCs were extracted for each window. The final feature map was a 13*N* × (100D − 2) matrix, where *N* = 1 when only MFCCs were extracted and *N* = 3 when deltas and delta–deltas were extracted as features. D is the duration length in second. Figure 3 represents MFCCs features from 1 s heart sound recording under both healthy and unhealthy conditions.

### 4.3. Model Interpretation

Another objective of this study is to compare the performance of deep learning models under similar conditions. Therefore, we built neural networks using different network structures, as shown in Figure 4. The overall design is MFCCs features input to the specific network module followed by a classification module to predict.

We applied three convolutional (Conv) layers with 32, 64, and 128 filters (size 3 × 3) on each CNN layer. Each Conv layer was connected to a batch normalisation layer (scale 1, offset 0, momentum rate 0.9) to speed up the training and reduce the sensitivity to network initialisation before the activation function (ReLU). Two max-pooling layers (size 2) were used before the second and third Conv layers to reduce the calculation amount. After the layers, the spatial features of the input were extracted and transferred to the fully-connected layer for classification.

The RNNs models used in this study were two layers. Because in our testing, the one-layer model did not perform well on the heart sound classification with approximately 75% accuracy (two layers performed around 90%). This testing result stands in line with the description in [31]. Furthermore, deeper layers did not show noticeable improvement in the performance as well. The number of hidden units was all set to 50 in each RNN layer, the state activation function was tanh and gate activation function was sigmoid.

### 4.4. Performance Metrics

Accuracy is the key parameter to evaluate the performance of a classification algorithm. However, the data structure in this study was not balanced (normal: abnormal is approximately 4:1), so we also calculated the true positive rate (sensitivity, *Se*), true negative rate (specificity, *Sp*), and overall score (*MAcc*).
(15)$$Acc=\frac{TP+TN}{TP+FP+TN+FN},\;Se=\frac{TP}{TP+FN},\;\;Sp=\frac{TN}{TN+FP},\;\;MAcc=\frac{Se+Sp}{2}$$
where *TP* (True positive) is the correctly classified healthy condition cases and *TN* (True negative) indicates the correctly classified unhealthy cases. Similarly, *FP* (False positive) represents the wrong detection on the normal sets and *FN* (False Negative) means incorrectly identified abnormal cases. The overall score (*MAcc*) is the average of the *Se* and *Sp*.

### 4.5. Training Settings

The optimiser selected in this study was stochastic gradient descent with momentum (SGDM, learn rate 0.01, momentum 0.9). Compared with the commonly used Adam optimiser, its convergence speed may be slower, but its convergent result can be better to find the best solution. The learning rate was constant 0.01, and data shuffling was conducted for each epoch. Max epoch was set to 100, but an early stopping was applied with the patience of 5 epochs to prevent overfitting.

### 4.6. Statistical Analysis

To statistically analyse the PCG duration effect and compare the performances of the models, we trained and tested all models ten times with different input PCG lengths to avoid random results. For each time, all the models were shared with the same random seed for the division of samples into training, validation, and testing sets to guarantee comparability of the results. A non-parametric test (Mann–Whitney U test) were conducted between results for statistical purposes.

## 5. Results

### 5.1. PCG Duration Effect on the Deep Learning Performance

The performances (in *MAcc*) of the proposed models using MFCCs, ∆MFCCs, and ∆²MFCCs as input are shown in Figure 5. The RNN models (LSTM, GRU, BiLSTM, and BiGRU) showed an apparent increase between 1 and 2 s (from approximately 0.87 to 0.89), whereas there was no evident change trend between 2 and 5 s. Taking BiLSTM data as an example for statistical analysis, the *p*-value of 1 and 2 s is 0.017, and it is over 0.45 among 2 to 5 s, which proved that our finding is not random. However, on the CNN model, the effect of PCG duration is negligible from our results (*p* > 0.3 between different durations). Thus, we can conclude that the 1 s length PCG segment is unsuitable for training RNN models to classify heart sound, but it is acceptable for CNN models.

### 5.2. RNNs vs. CNN

Because there is no apparent PCG duration effect shown in Figure 5 between 2 and 5s, in this part, we analyse the performances of all the proposed models based on 5 s PCG duration, which are summarised in Table 3. RNN models outperformed the CNN model with the higher average accuracy, sensitivity, and overall score. Furthermore, the difference among the four RNN models was negligible within 1.5% (*p* ≥ 0.175). The best performance came from GRU with 94.07% in *Acc*, 94.81% in *Sp*, 91.29% in *Se*, and 93.05% in *MAcc*. According to the current result, it is hard to say which RNN model is the best for heart sound classification. Notably, RNNs had better overall performance than the CNN model using MFCCs as input (*p* ≤ 0.044 paired with RNN models).

### 5.3. Effect of Using ∆MFCCs and ∆²MFCCs as Features

The performance of adding ∆MFCCs and ∆²MFCCs as features are shown in Figure 6. Because the result is applicable on all RNN models, hereby in Figure 6, only BiLSTM performance is selected to display. It indicates no observed improvement on the RNN models by using extended MFCCs as the feature (*p* = 0.748). However, on the CNN model, using ∆MFCCs and ∆²MFCCs increased the classification accuracy a little bit (≤2.5%, *p* = 0.003).

## 6. Discussion

This study analyses the PCG duration effect on the heart sound classification performance and compares the deep learning models using MFCCs as input features. Because of the machine learning algorithm design, when limited training data are given, it is a standard practice to segment the database further to increase the number of training sets for accuracy improvement. Conventional segmentation (identification) breaks the PCG signals into each heart cycle for heart sound classification. However, segmentation is complicated work and cannot always complete the task accurately, especially when there is a murmur or noise inside. More studies just clipped the datasets by seconds in recent years, but no detailed analysis on the effect of the split length. In this study, the results have shown that 1 s length PCG is not an appropriate length on RNN models, while it might be applicable for the CNN model. A normal heartbeat at rest ranges from 60 to 100 per minute, which means 1 s length can only cover a complete heart cycle. For RNNs, they are specialised in processing sequence data. Thus, the whole heart cycle will be helpful to provide more comprehensive information for the trained network. However, the CNN model has the advantage of exploring specific spatial characteristics, which means missing partial information, mainly the edge information, may not directly affect the final performance of the classification. From 2 to 5 s, there is a balance between the information amount of single data and total sample size that did not affect the classification results. As a result, we suggest using 2 s PCG length to process the datasets on RNN models because a shorter segment means the potential for more repeated testing on one testing recording. This may be helpful to reduce the random error increasing the algorithm robustness, also it could control the sensitivity and specificity by appropriate threshold.

The comparison between deep learning models showed that RNNs performed better than the CNN model when using MFCCs as input. Because the MFCCs as a sequence to describe the instantaneous, spectral envelope shape of the heart sound signal did not hold too much spatial information for CNN to extract. The CNN model may perform better with a deeper structure or analyse the pattern-based time-frequency features such as the heart sound spectrogram, Mel-frequency cepstrum, etc. However, it will also require greater computing power to process during training. Among the RNN models, though their performances were quite close in our study, we suggest GRU based on the computing amount and processing time. In our testing (CPU: Intel(R) ES 1650 v3 @3.50GHz, GPU: NVIDIA Quadro M2000), using GRU to finish the training (5 s, MFCCs only, 30 epochs) costs approximately 395 s, while LSTM needed 433 s and BiLSTM spent 588 s. From the structure, GRU is simpler than LSTM with fewer gates to calculate, bidirectional RNNs almost double the computing load by calculating the inverse propagation. Therefore, a more straightforward structure neural unit was recommended when no noticeable performance improvements were shown among them.

Thirdly, in exploring using ∆MFCCs and ∆²MFCCs as additional features, we found no improvement on the final classification results of RNN models, but a slight increase in CNN models. This may be because ∆MFCCs and ∆²MFCCs are derivatives of MFCCs, for RNN input sequence, no additional information was added. However, the CNN model viewed the input as a pattern; extended features can supplement more spatial characteristics. Another interesting finding is that most MFCCs information to classify heart sound as normal or abnormal was based on the first three dimensions (MFCC1, MFCC2, and MFCC3), as shown in Figure 3. These dimensions correspond to the low-frequency band part of the signal. We tested using the first three MFCCs instead of all 13 MFCCs as input and found *Acc* 91.43%, *Sp* 94.39%, *Se* 82.07%, and *MAcc* 88.23% on 5 s BiLSTM, which did not show an apparent decrease in the performance. Thus, it is a potential way to reduce the calculation amount.

## 7. Conclusions

This study found that the PCG duration will affect the deep learning performance, and the commonly used 1 s length is not a reasonable option to process the datasets. We suggested starting from 2 s since a bit longer duration can provide more information and benefit the classification performance. However, only increasing the input length without changing network architecture does not guarantee better performance. When using MFCCs as training features, RNNs outperformed the CNN model, whereas there is no apparent difference among the RNN models (LSTM, BiLSTM, GRU, or BiGRU, within 1.5%). In comparison, GRU has the advantages of a smaller computational load and a faster training speed. For MFCC features, adding dynamic information (∆MFCCs and ∆²MFCCs) of the heart sound did not improve the RNN performance, and the improvement on CNN is also minimal.

## Figures and Tables

**Figure 1 sensors-22-02261-f001:**
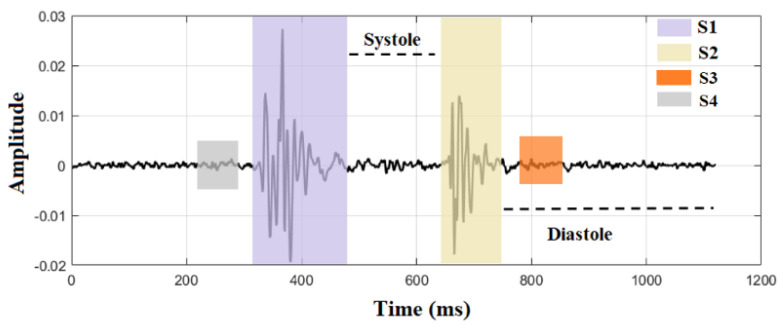
Visualisation of heart sound signal with its component locations.

**Figure 2 sensors-22-02261-f002:**
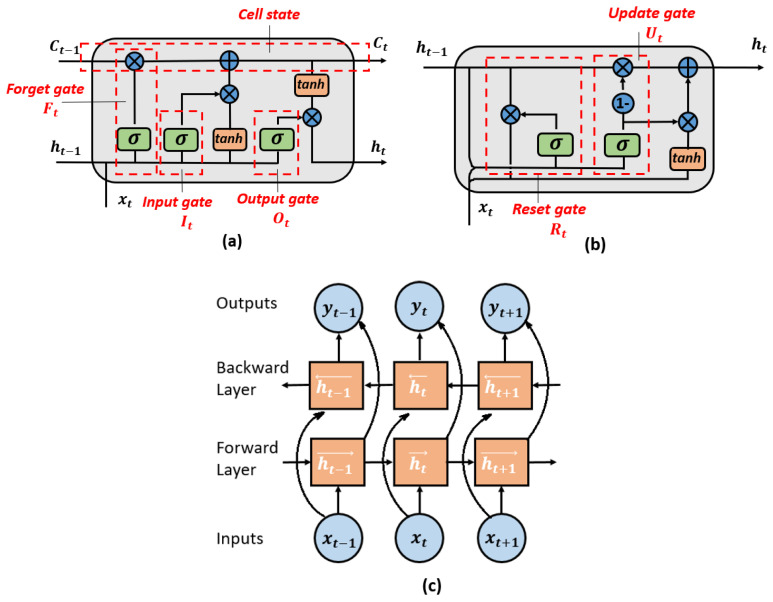
Graphical structure of (**a**) LSTM unit; (**b**) GRU; (**c**) BRNN.

**Figure 3 sensors-22-02261-f003:**
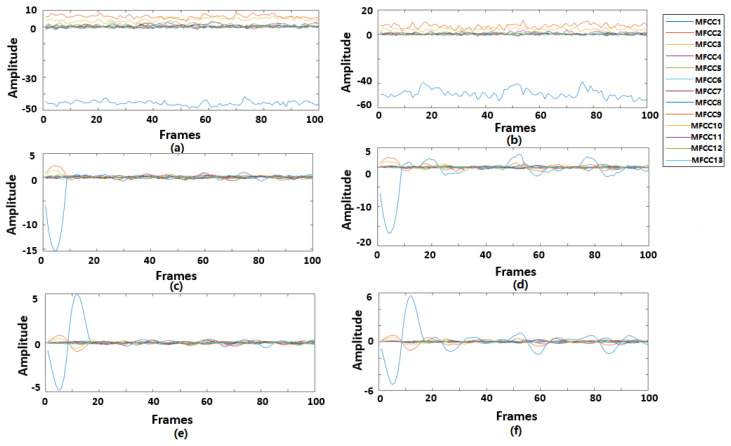
(**a**) MFCCs, (**c**) ∆MFCCs, (**e**) ∆²MFCCs for normal heart sound recording. (**b**) MFCCs, (**d**) ∆MFCCs, (**f**) ∆²MFCCs for abnormal heart sound recording.

**Figure 4 sensors-22-02261-f004:**
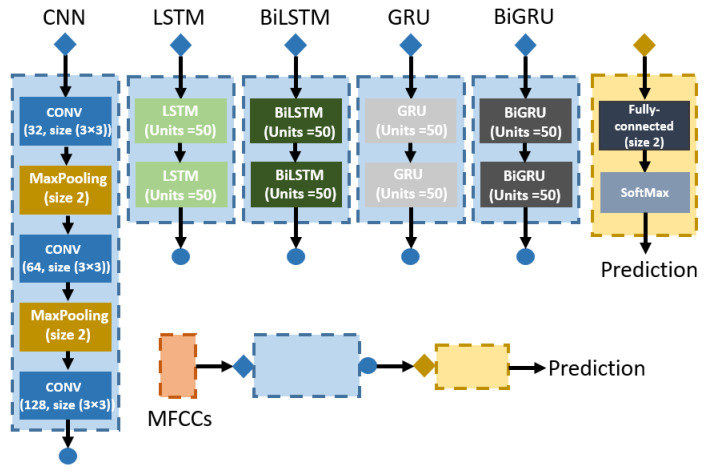
The deep learning network models used in the study: 3-layer CNN model, 2-layer LSTM, BiLSTM, GRU, and BiGRU models.

**Figure 5 sensors-22-02261-f005:**
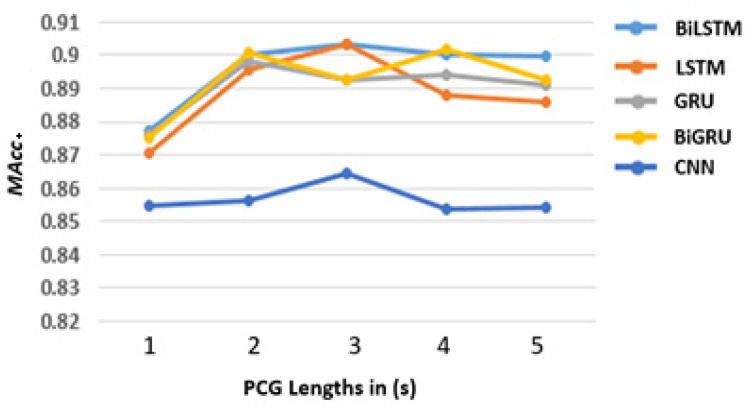
The proposed models’ performance (10 times average) with different input PCG signal lengths.

**Figure 6 sensors-22-02261-f006:**
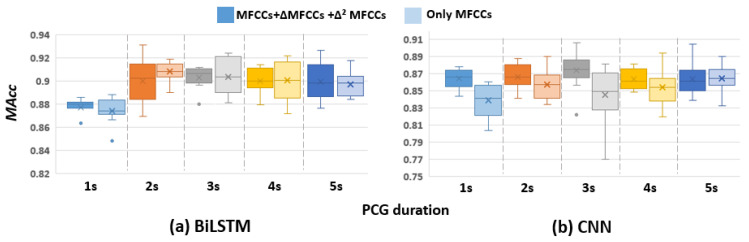
Comparison between using MFCCs only and using MFCCs with its deltas and delta–deltas on BiLSTM and CNN.

**Table 1 sensors-22-02261-t001:** Heart Sound Components and their properties.

Heart Sound	S1	S2	S3	S4
Duration (ms)	100–160	80–140	40–50	40–50
Frequency (Hz)	30–50	40–70	<30	<20
Occurrence	Sound of mitral and tricuspid valve closure	Sound of aortic and pulmonic valve closure	The sound caused by an increase in ventricular blood volume	Sound of an atrial gallop produced by blood being forced into a stiff ventricle

**Table 2 sensors-22-02261-t002:** Recent advancements in heart anomaly detection using deep learning.

Authors	Year	Segmented (Input Length)	Features	Model	*Acc* (%)
Huai et al. [10]	2020	No. (5 s intervals, 2 s window)	Time-Frequency (Spectrogram)	CNN + LSTM	91.06
Deng et al. [13]	2020	No. (5 s)	MFCCs, ΔMFCC, Δ²MFCC	CNN + RNN	98.34
Xiao et al. [30]	2020	No. (3 s length, 1 s shift)	Raw signals, MFCCs, PSDs	CNN	93
Dissanayake et al. [12]	2020	No. (1 s, 0.1 s shift)	MFCCs	LSTM, CNN	99.72
Zhang et al. [6]	2019	No. (2 s)	Temporal Quasi-Periodic Features	LSTM	94.66
Latif et al. [31]	2018	Yes. (2, 5, and 8 cycles)	MFCCs	RNNs	98.61
Maknickas and Maknickas [9]	2017	No. (128 × 128 frames)	MFCCs	CNN	84.15
Rubin et al. [32]	2017	Yes. (3 s)	Time-Frequency, MFCCs	CNN	84

**Table 3 sensors-22-02261-t003:** Comparison between the deep learning models with 5 s PCG input (10 times, average ± standard deviation %).

Model	*Acc*	*Sp*	*Se*	*MAcc*
LSTM	91.86 ± 1.20	**95.42 ± 1.45**	81.75 ± 5.95	88.58 ± 2.44
BiLSTM	**92.64 ± 0.75**	95.14 ± 1.33	**84.77 ± 3.87**	**89.95 ± 1.58**
GRU	92.13 ± 0.44	95.22 ± 1.37	83.01 ± 2.69	89.12 ± 0.87
BiGRU	92.35 ± 0.72	95.31 ± 1.87	83.24 ± 3.46	89.27 ± 1.04
CNN	90.08 ± 1.22	93.80 ± 2.46	79.02 ± 4.57	86.41 ± 1.7

## Data Availability

Data sharing not applicable.

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
