# Peer review of "The Effect of Signal Duration on the Classification of Heart Sounds: A Deep Learning Approach"

_sensors, 2022, doi:10.3390/s22062261_

Round 1

Reviewer 1 Report

The authors focus on the issue of signal length and its impact on the classification of heart sounds. The topic as such is interesting and important. However, the manuscript needs some improvements and clarifications.

The related works and general description of proposed methods is good.

It is not clear from the description whether exactly the same samples were used for all classification methods in the declared division into training, validation and testing sets in order to guarantee comparability of the results.

Please state explicitly that classification into 2 classes was performed. It is "hidden" in data description that normal and abnormal cases were used. This important information must be clearly and explicitly written.

More detailed information must be added to the description of each used neural network.

The sections 4.5 and 4.6 are too brief. Description in 4.6 is not clear at all. It is written that all models were trained and tested ten times with different input PCG lengths. Does it mean that with each PCG length (1s, 2s,..5s) training and testing was performed ten times? How were the data samples divided into training, testing and validation sets? This is crucial for understanding the whole proposed approach and reached results.

Language needs revision. There are some grammatical and mistyping errors and unclear formulations. For example, line 361 sentence: "Because ten times cannot ensure the results are normally distributed." Please try to formulate the idea clearly.

Author Response

Thank you very much. We have uploaded the response in a file. 

Reviewer 2 Report

This manuscript mainly focuses on analyzing the duration effect on the commonly used deep learning methods to provide insight for future studies in data processing, classifier, and feature selection. Some improvements are needed before being published on Sensors.

  1. It’s better to improve the writing of this manuscript to meet the standard of scientific papers.
  2. What’s the accuracy of the method proposed in this manuscript?
  3. The Table captions doesn’t match with the text in the manuscript.
  4. There are some spell mistakes in the manuscript.
  5. The format of some references should be revised.
  6. The figures can be better organized. Especially the labels in the figures should be consistent. The permission should be got if the figures are from other papers.
  7. What’s the meaning of “?” in Table 2?
  8. What about the Training Efficiency of the method? It's better to clarify. 
  9. For the conclusion, it's not a clear summary of the content and better to be improved. 

Author Response

(The authors gave the same response as above.)

Round 2

Reviewer 1 Report

The manuscript has been significantly improved. The authors responded to all questions and comments.